# Changes in Human Population Density and Protected Areas in Terrestrial Global Biodiversity Hotspots, 1995–2015

**Caitlin Cunningham [1],* and Karen F. Beazley [2]**

[1]   Interdisciplinary PhD Programme, Dalhousie University, Halifax, NS B3H 4R2, Canada
[2]   School for Resource and Environmental Studies, Dalhousie University, Halifax, NS B3H 4R2, Canada; karen.beazley@dal.ca
*    Correspondence: caitlin.cunningham@dal.ca; Tel.: +1-902-412-2732

**Abstract:** Biodiversity hotspots are rich in endemic species and threatened by anthropogenic influences and, thus, considered priorities for conservation. In this study, conservation achievements in 36 global biodiversity hotspots (25 identified in 1988, 10 added in 2011, and one in 2016) were evaluated in relation to changes in human population density and protected area coverage between 1995 and 2015. Population densities were compared against 1995 global averages, and percentages of protected area coverage were compared against area-based targets outlined in Aichi target 11 of the Convention on Biological Diversity (17% by 2020) and calls for half Earth (50%). The two factors (average population density and percent protected area coverage) for each hotspot were then plotted to evaluate relative levels of threat to biodiversity conservation. Average population densities in biodiversity hotspots increased by 36% over the 20-year period, and were double the global average. The protected area target of 17% is achieved in 19 of the 36 hotspots; the 17 hotspots where this target has not been met are economically disadvantaged areas as defined by Gross Domestic Product. In 2015, there are seven fewer hotspots (22 in 1995; 15 in 2015) in the highest threat category (i.e., population density exceeding global average, and protected area coverage less than 17%). In the lowest threat category (i.e., population density below the global average, and a protected area coverage of 17% or more), there are two additional hotspots in 2015 as compared to 1995, attributable to gains in protected area. Only two hotspots achieve a target of 50% protection. Although conservation progress has been made in most global biodiversity hotspots, additional efforts are needed to slow and/or reduce population density and achieve protected area targets. Such conservation efforts are likely to require more coordinated and collaborative initiatives, attention to biodiversity objectives beyond protected areas, and support from the global community.

**Keywords:** biodiversity conservation targets; threat assessment; prioritization; biodiversity hotspots; human population density; protected areas

## 1. Introduction

Anthropogenic activities have generated significant negative impacts on natural environments. No other species has exerted such immense influence on all the other species, and rapid growth of human populations poses additional threats to biodiversity [1–3]. Areas with high human population densities are particularly threatened, through direct and indirect effects [4–7]. To help counter such threats, biodiversity conservation has emerged as a global priority, to be addressed at multiple scales, and including tactics such as the establishment of protected areas and other effective area-based conservation measures [8,9]. Since biodiversity and threats to it are not evenly distributed across the

planet, various frameworks for identifying global priorities for conservation have been developed [8]. Conservation priorities are often dictated through frameworks based on vulnerability (relative risk or threat to biodiversity) and irreplaceability (the degree to which options for safeguarding biodiversity values elsewhere are limited or restricted) [8,10–12]. These two measures have been combined in a variety of ways on the global scale to identify conservation priorities, including crisis ecoregions [13], endemic bird areas [14], centers of plant diversity [15], megadiversity countries [16], global 200 ecoregions [17,18], high-biodiversity wilderness areas [19], frontier forests [20], and last of the wild [21]. In a review of global prioritization frameworks, Brooks et al. [8] found that most frameworks prioritized irreplaceability, but that some were 'reactive' (prioritizing high vulnerability) and others 'proactive' (prioritizing low vulnerability). In this study, we focus on global biodiversity hotspots [12–15,17–19]. Biodiversity hotspots meet two key criteria: (i) a minimum of 1500 endemic plant species; and, (ii) a loss of at least 70% of original habitat extent [11]. By this definition they represent a reactive approach to conservation prioritization. They are considered high priorities for conservation attention because of their importance (irreplaceability) in safe-guarding global biodiversity and the threats (vulnerability) they are under.

Global biodiversity hotspots were first identified by Myers [12]. Initially, 10 forest biodiversity hotspots were identified, based on endemic species richness and threats from human activities, with eight hotspots from other ecosystem types added later [22]. An extensive systematic global review subsequently identified seven additional global biodiversity hotspots, for a total of 25 [23,24]. Cincotta et al. [25] published 1995 human population density and growth rates for these 25 biodiversity hotspots, providing an important methodology and baseline study for assessing future levels of these measures in these and other conservation priority areas. The earlier biodiversity hotspot frameworks have been revised several times since their inception. A second systematic global update [26] redefined several hotspots and added others. In 2011, an additional hotspot was identified [27], bringing the total number of global hotspots to 35 [11]. In 2016, the global analysis was updated, bringing the total number to 36 [28]. The 25 hotspots originally identified by Myers, Mittermeier, and colleagues [12,24] remain in the updated set of 36 hotspots, though some have been renamed. Collectively, the global biodiversity hotspots cover a tiny fraction of the planet—just 4.6% of the Earth's total surface area (15.4% of the land area)—but they are important to biodiversity, representing 50% of vascular plants and 42% of terrestrial vertebrates as endemics [11,26].

The Strategic Plan for Biodiversity 2011–2020, signed at COP10, outlines global targets for biodiversity conservation. One of the targets in the plan, Aichi target 11, specifies that "by 2020, at least 17% of terrestrial and inland water areas . . . , especially areas of particular importance for biodiversity and ecosystem services, are conserved through effectively and equitably managed, ecologically representative and well-connected systems of protected areas and other effective area-based conservation measures" [29]. This minimum global target of 17% is politically defined rather than scientifically defensible and, thus, is regarded by many as an interim target [30]. Most scientific studies estimate 25–75% of a region is needed to protect biodiversity, some are calling for 50% (half earth), and new global targets for beyond 2020 are currently being negotiated. Furthermore, the best practice in conservation planning calls for higher than average percentages of protection for rare or geographically-limited remnants of hotspots of diversity and rarity [30–33]. Given that the irreplaceable habitats represented in the global biodiversity hotspots have already been reduced to approximately 14.5% of their original extent, from 23.5 million km$^2$ to 3.4 million km$^2$ in 2004 [11,26], strong argument could be made for protecting as much as 100% of their remaining intact ecosystems.

In light of the irreplaceability and vulnerability of these global biodiversity hotspots, we analyze how well their conservation is currently being achieved relative to measures in 1995, in comparison to protected area targets of 17% and 50%, and in the face of one of the biggest threats to biodiversity—increasing human population density (Figure 1), which presents additional challenges for conservation planning [3,25,34–36]. Our assessment follows up on Cincotta et al.'s [25] similar study in 1995, providing an update and comparison, and complements a recent assessment by

Weinzettel et al. [37] of conservation priorities among hotspots based on a footprint measured as loss of potential net primary production due to agricultural production and final consumption. Such diverse and on-going analyses are important in moving forward with future conservation strategies, such as new protected area targets beyond 2020. They also serve to illustrate the deficiencies inherent in relying upon discrete surrogate measures of threat no matter how well established, the ineffectual nature of global assessments to direct or inform within site or local conservation initiatives, and the importance of collaborative international and local conservation approaches, including those that extend beyond protected areas.

## 2. Methods

Quantitative analyses of human population densities (1995, 2015, and projected to 2020) and percentages of protected area coverage (1995 and 2015) were conducted for 36 global biodiversity hotspots identified by Mittermeier and colleagues [11] and the Critical Ecosystem Partnership Fund [28] (Figure 1). Changes in population density and protected area coverage from 1995 to 2015 were calculated. Population densities were compared against global averages, and percentages of protected area coverage were compared against area-based targets outlined in Aichi target 11 (17% by 2020) and half Earth (50%). Average population density and percentage of protected area coverage for each hotspot were then plotted against each other to evaluate relative levels of threat to biodiversity conservation in 1995 and 2015. Changes from 1995 to 2015 in the numbers and suites of global biodiversity hotspots falling within the high threat category (i.e., higher than global average population densities, and lower than target percentages for protected areas) were assessed.

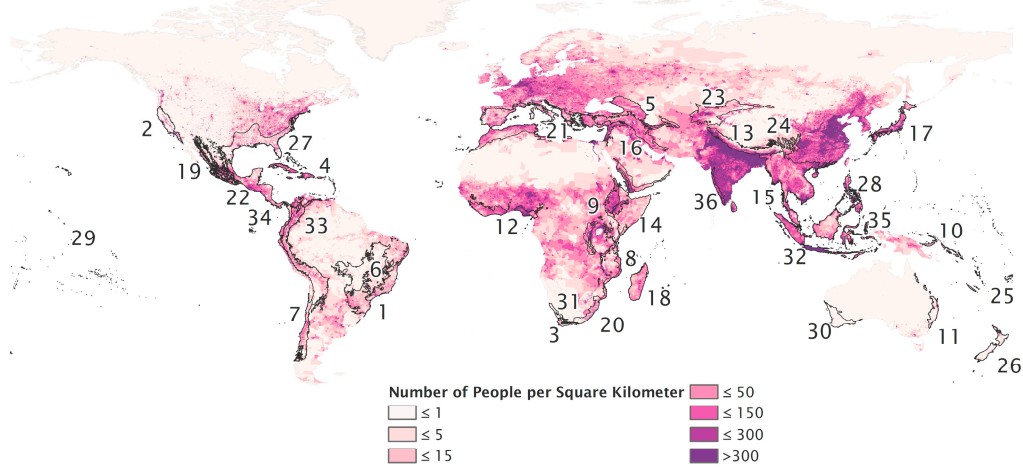

**Figure 1.** Map showing the human population density (number of people per square kilometer) and the location of the biodiversity hotspots. Hotspots: (1) Atlantic Forest, (2) California Floristic Province, (3) Cape Floristic Region, (4) Caribbean Islands, (5) Caucasus, (6) Cerrado, (7) Chilean Winter Rainfall and Valdivian Forests, (8) Coastal Forests of Eastern Africa, (9) East Melanesian Islands, (10) Eastern Afromontane, (11) Forests of East Australia, (12) Guinean Forests of West Africa, (13) Himalaya, (14) Horn of Africa, (15) Indo-Burma, (16) Irano-Anatolian, (17) Japan, (18) Madagascar and the Indian Ocean Islands, (19) Madrean Pine-Oak Woodlands, (20) Maputaland-Pondoland-Albany, (21) Mediterranean Basin, (22) Mesoamerica, (23) Mountains of Central Asia, (24) Mountains of Southwest China, (25) New Caledonia, (26) New Zealand, (27) North American Coastal Plain, (28) Philippines, (29) Polynesia-Micronesia, (30) Southwest Australia, (31) Succulent Karoo, (32) Sundaland, (33) Tropical Andes, (34) Tumbes-Choco-Magdalena, (35) Wallacea, and (36) Western Ghats and Sri Lanka.

### 2.1. Population Density

Population density methods followed Cincotta et al. [25]. Average human population density for each biodiversity hotspot was calculated using the *Gridded Population of the World* data for 1995, 2015, and 2020 (predicted) [38]. These data were collated by the Socioeconomic Data and Applications Center at NASA using national censuses and population registers around the world at a 30 arc-second (approximately one kilometer at the equator) resolution. The authors of the data note several limitations of the data, such as the extrapolation of census data to the specific year of the data and the mapping of the geographical boundaries themselves [39]. The data are therefore not appropriate for use at small, localized scales, but are useful for analyses such as ours that focus on broad trends over large areas. Changes in average population density between 1995 and 2015, as well as 2015 and 2020, were calculated via raster calculation. The change in population density for each hotspot was compared to the change in global (across all land, excluding ice or rock covered land; consistent with [25]) average density to identify those biodiversity hotspots in which the average population density is increasing at rates faster than the global average.

### 2.2. Percent Protected Area

Protected areas data for 1995 and 2015 were obtained from the World Database of Protected Areas (WDPA), United Nations Environment Programme (UNEP) [40]. While most of these data are in polygon form, about 9% of protected areas in this database are recorded as points. Following recommendations in the user manual [41], points were converted to polygons using buffers based on the reported size of the protected area in the database. The dissolve tool was then used to eliminate the double counting of areas of overlap. These processed data were then used to determine the percentage of terrestrial protected area in each biodiversity hotspot in 1995 and 2015. The change in percentage area protected in each hotspot between 1995 and 2015 was also calculated and compared to the global change in percentage of terrestrial protected area over the same time period.

### 2.3. Combining Population Density and Percent Protected Area to Assess Threat Level

Pearson correlation coefficients were calculated for average population density and percentage of protected area for each biodiversity hotspot. The two factors were then used to determine the level of threat associated with each biodiversity hotspot, specifically: (i) whether average population density in 2015, change in average population density from 1995 to 2015, and predicted change in average population density for 2015 to 2020 were above or below the global averages; and, (ii) whether the 17% and 50% protected areas targets were met. The results for each biodiversity hotspot at each time period were plotted, with population density values on the y-axis, and percent protected area on the x-axis. Results were grouped into quadrants defined by average population densities above or below the global average in 1995 and whether or not the Aichi target 11 goal of 17% terrestrial protection has been met.

## 3. Results

### 3.1. Population Density

Over the 20-year time frame, average population density across all biodiversity hotspots increased by 36%, from 76 people per square kilometer (ppl/km$^2$) to 103 ppl/km2, which was a smaller increase in density than for the world as a whole (47% increase; from 38 to 56 ppl/km$^2$) (Table A1). However, 11 hotspots saw average population density increases at rates higher than the global average. The highest *raw* change was found in the Philippines (28) (103 ppl/km$^2$); the highest *rate* of change was found in the Horn of Africa (14) (88.24%). In both 1995 and 2015, the average population density across all biodiversity hotspots was about double the global average in both 1995 (76 ppl/km$^2$ and 38 ppl/km$^2$, respectively) and 2015 (103 people/km$^2$; 56 ppl/km$^2$) (Figures 2 and 3). In 2015, average population

densities ranged from 3 ppl/km$^2$ in the Succulent Karoo (31) to 345 ppl/km$^2$ in the Philippines (28); 21 of the 35 biodiversity hotspots had average population densities above the global average.

Overall, the predicted average population density for 2020 in hotspots is 112 ppl/km$^2$, compared to a predicted global average of 61 ppl/km$^2$. Average population densities in biodiversity hotspots are predicted to range from 4 ppl/km$^2$ in the Succulent Karoo (31) to 378 ppl/km$^2$ in the Philippines (28). In total, 21 of the 35 biodiversity hotspots are expected to have average population density higher than the global average (the same as the 2015 data). This predicted change in average population density represents an 8.74% increase in the biodiversity hotspots, which is on par with the world, which is predicted to see an 8.93% increase. The full results for the population densities in biodiversity hotspots for 1995, 2015, 2020 (predicted), and the changes between them can be found in Table A1.

### 3.2. Percent Protected Area

In 2015, 23.22% of the terrestrial and inland water areas of biodiversity hotspots was protected, which was above the global total of 14.7% (Table A2). However, there is high variability in the percentage protected of each hotspot, ranging from just 2.19% in the East Melanesian Islands (9) to 57.74% in New Caledonia (25). Overall, 23 hotspots were protecting more than the global average of 14.7% and 19 had met or exceeded the Aichi target 11 goal of 17% protected area. Two hotspots (Cerrado (6) and New Caledonia (25)) exceeded 50% protection. From 1995 to 2015, the protected area within hotspots increased by 12.44% on average, which was higher than the global increase of 4.9% over the same period. In 23 of the 36 biodiversity hotspots, the percentage of land in protected areas increased by more than the global average. The highest increases were seen in Cerrado (6) (36.86%) and the lowest increases were seen in the Horn of Africa (14) (0.46%). The full results for the percentage of area protected in each biodiversity hotspot for 1995 and 2015 and the change between them can be found in Table A2.

### 3.3. Comparing Population Density and Percent Protected Area

The average human population density and percentage of protected area in the biodiversity hotspots is not statistically correlated for 1995 (correlation coefficient 0.30) or for 2015 (correlation coefficient −0.09) (Table A3). It is worth noting, however, that three (New Caledonia (25), Cerrado (6) and Tropical Andes (33)) of the four hotspots with the highest percentages (>45%) of protected area in 2015 also had relatively low average population densities; among the four, only Cape Floristic Region (3) was above the global average. The Aichi Target 11 aim of 17% protection was met in hotspots with a range of average population densities, from high densities in Japan (17) (29.42% protected; 336 ppl/km$^2$ average population density) to low densities in Southwest Australia (30) (17.06% protected; 6 ppl/km$^2$ average population density).

The mean population density and percentage terrestrial area protected were plotted against each other for both 1995 (Figure 2) and 2015 (Figure 3). This placed hotspots into quadrants based on whether or not mean population densities were above the 1995 and 2015 global averages and whether or not the 17% or 50% targets for terrestrial protected area were met. In 1995, there were only three biodiversity hotspots with 17% or more of terrestrial area protected and a mean population density at or below the 1995 global average (Figure 2; Table 1). In 2015, there were five hotspots in this quadrant: the three from 1995 ((Cerrado (6), New Caledonia (25) and New Zealand (26)), plus Southwest Australia (30) and Succulent Karoo (31) (Figure 3; Table 1). If the 2015 global average population density is used, an additional three hotspots would be in this lower-threat group (Tropical Andes (33), Chilean Winter Rainfall and Valdivian Forests (7) and Forests of East Africa (11)). No hotspots met or exceeded a 50% protected area target in 1995, however two (Cerrado (6) and New Caledonia (25)) met the target in 2015, representing the least-threatened hotspots.

In 1995, there were 22 hotspots in the highest-threat quadrant, with mean population densities above the global average and less than 17% protected area (Table 1). In 2015, there were seven fewer hotspots in this quadrant due to increases in protected area, and one due to lower average population

density (Mountains of Southwest China (24)), leaving 14 hotspots in the high threat quadrant. If 50% is considered as a protected area target, all but two hotspots (Cerrado (6) and New Caledonia (25)) shift into the higher threat category (mean population densities above the global average and less than 50% protected area).

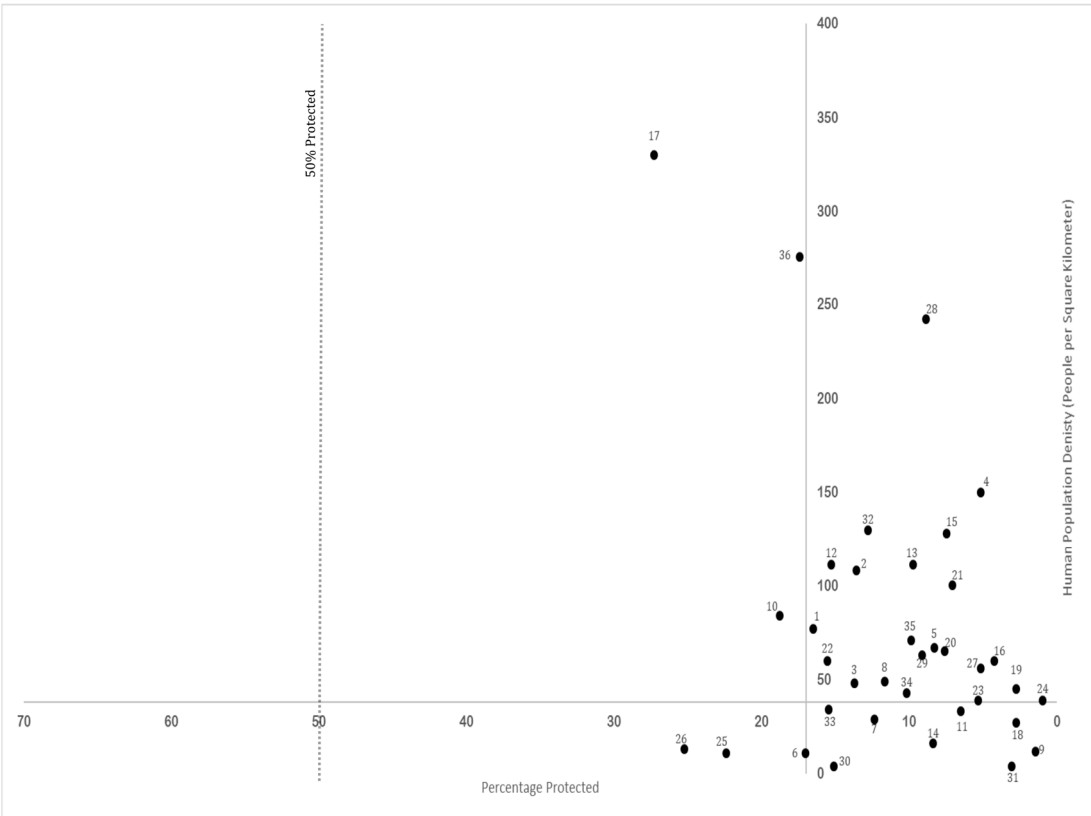

**Figure 2.** 1995 mean population densities (number of people per square kilometer) and percentage terrestrial area protected in each biodiversity hotspot. The x-axis crosses the y-axis at 38 people per square kilometer (the global average in 1995). The y-axis crosses the x-axis at 17% protected area, the target defined in Aichi target 11; the half-earth target of 50% protection is also shown. The upper-right quadrant represents hotspots that are under the most threat—they have an average human population density above the 1995 global average and less than 17% protected area. Lower-left quadrants represent hotspots under lesser threats—they have average human population densities at or below the global average and at least 17% of the area is under protection, or at least 50% protection (the lowest threat category). Hotspot numbers correspond with Figure 1.

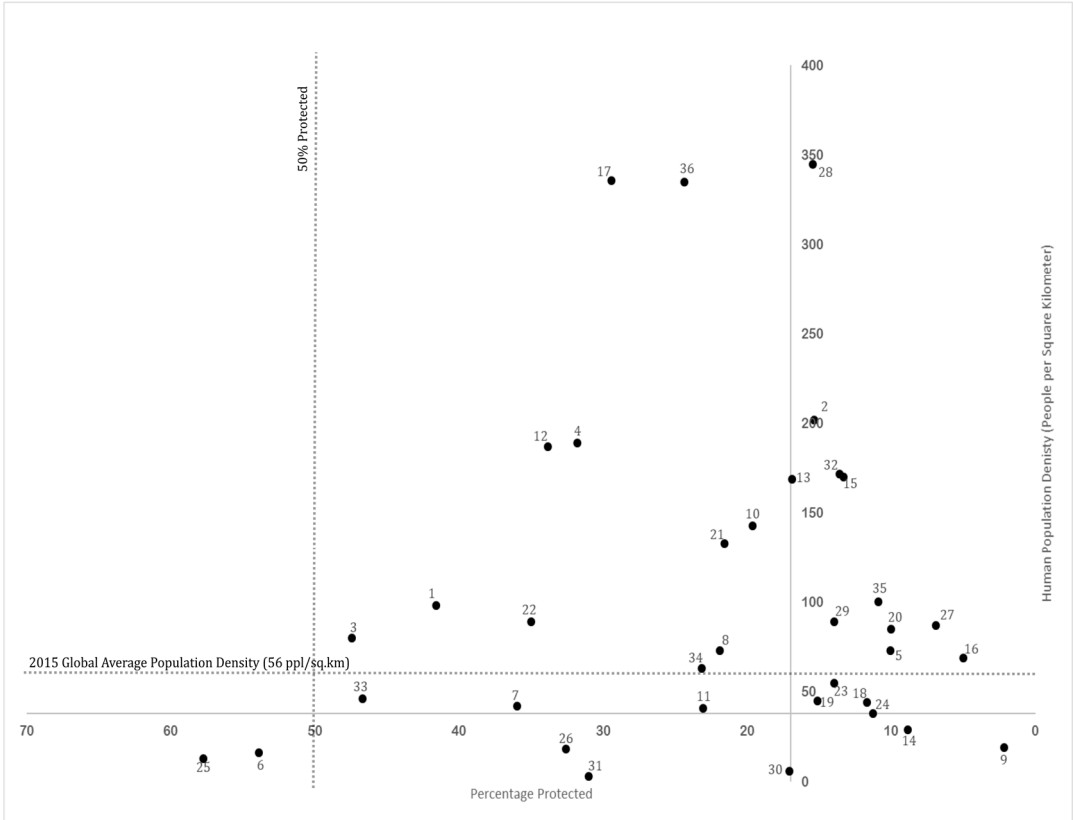

**Figure 3.** 2015 mean population densities (number of people per square kilometer) and percentage terrestrial area protected in each biodiversity hotspot. The x-axis crosses the y-axis at 38 people per square kilometer (the global average in 1995); the 2015 global average population density (56 people per square kilometer) is also indicated. The y-axis crosses the x-axis at 17% protected area coverage, the target under Aichi target 11; the half-earth target of 50% protection is also indicated. The upper-right quadrant represents hotspots that are under the most threat—they have an average human population density above the 2015 global average and less than 17% protected area. The lower-left quadrant represent hotspots under the least threat—they have an average human population density at or below the 1995 global average and at least 50% protected area. Hotspot numbers correspond with Figure 1.

**Table 1.** Biodiversity hotspots falling within high- and low-threat categories in analyses based on average human population densities relative to global average and protected area relative to 17% and 50% targets in 1995 and 2015, as shown in Figures 2 and 3.

| 1995 | 2015 |
|---|---|
| **High Threat Categories** **(Average human population density >1995 global average; <17% protected area)** | |
| Atlantic Forest (1) California Floristic Province (2) Cape Floristic Region (3) Caribbean Islands (4) Caucasus (5) Coastal Forests of Eastern Africa (8) Guinean Forests of West Africa (12) Himalaya (13) Indo-Burma (15) Irano-Anatolian (16) Madrean Pine-Oak Woodlands (19) Maputaland-Pondoland-Albany (20) Mediterranean Basin (21) Mesoamerica (22) Mountains of Central Asia (23) Mountains of Southwest China (24) North American Coastal Plain (27) Philippines (28) Polynesia-Micronesia (29) Sundaland (32) Tumbes-Choco-Magdalena (34) Wallacea (35) | Average human population density >2015 and >1995 global averages; <17% protected area: California Floristic Province (2) Caucasus (5) Himalaya (13) Indo-Burma (15) Irano-Anatolian (16) Maputaland-Pondoland-Albany (20) North American Coastal Plain (27) Philippines (28) Polynesia-Micronesia (29) Sundaland (32) Wallacea (35) Average human population density >1995–2015 global average; <17% protected area: Madagascar and the Indian Ocean Islands (18) Madrean Pine-Oak Woodlands (19) Mountains of Central Asia (23) |
| **Low threat categories** **(Average human population density ≤1995 global average; ≥17% protected area)** | |
| (Average human population density ≤1995 global average; ≥17%–<50% protected area: | |
| Cerrado (6) New Caledonia (25) New Zealand (26) | Cerrado (6) New Caledonia (25) New Zealand (26) Southwest Australia (30) Succulent Karoo (31) |
| (Average human population density ≤1995 global average; ≥50% protected area: | |
| | Cerrado (6) New Caledonia (25) |

## 4. Discussion

### 4.1. Using Human Population Density as a Measure of Threat to Biodiversity

Cincotta and colleagues [25] examined mean human population densities (based on the same 1995 data used in this paper) and population growth rates between 1995 and 2000 in 25 biodiversity hotspots. For these 25 hotspots, our calculations of 1995 mean human population densities yielded similar results (any differences were non-significant and likely attributable to the different underlying hotspot datasets used). Although Cincotta and colleagues also looked at population growth rates in these hotspots, we analysed actual (1995–2015) changes in population density because population growth rates can be misleading indicators of risk [25]. These different methods yielded different pictures of the changes in human populations in these hotspots. For example, in our analysis, the California Floristic Province (2) was identified as having the fastest growing population density (an 87% increase between 1995 and 2015, as compared to global average of 47%), whereas Cincotta et al. [25] calculated the population

growth rate for this hotspot as right at the global average (between 1995 and 2000), which was a relatively low growth rate as compared to other hotspots.

Although human population density is frequently used as a proxy for threat to biodiversity [3,25,42–44], there are caveats to the method. First, it only measures human presence in the discreet analytical unit and does not take into account the indirect effects that localized high densities, such as urban populations, can have on surrounding ecosystems. Cities can alter ecosystems over great distances, potentially even thousands of kilometers away. Second, using average population density throughout a biodiversity hotspot obscures patterns of population distribution within the area. The hotspots are mostly large with population density widely variable within them. For example, the Mountains of Southwest China (24) and Forests of Eastern Australia (11) have similar average population densities (38 and 41 ppl/km$^2$, respectively) and total areas (262,129 and 254,388 km$^2$, respectively), but very different patterns of population distribution. In the Mountains of Southwest China (24), the population is spread throughout the hotspot (Figure 4a). Conversely, the human population of the Forests of Eastern Australia (11) is concentrated along the coast in a few, densely populated cities: Sydney, Brisbane, Newcastle and Cairns. The rest of the hotspot is virtually uninhabited (Figure 4b). Obscuring the pattern of population distribution within hotpots also obscures the ecological footprint of the various populations that live throughout biodiversity hotspots. In hotspots where population is distributed throughout, conservation initiatives may be more prevalent throughout the region in a 'land-sharing' approach, in which humans and nature coexist [45–47]. In those where the population is more spatially concentrated, greater opportunities may exists for 'land-sparing', in which larger and more connected protected areas may be set aside. Further, it is well established that land-use activities such as agriculture, timber harvesting and mining can be extensive and intensive in terms of impact on biodiversity, even in areas with low population densities [37].

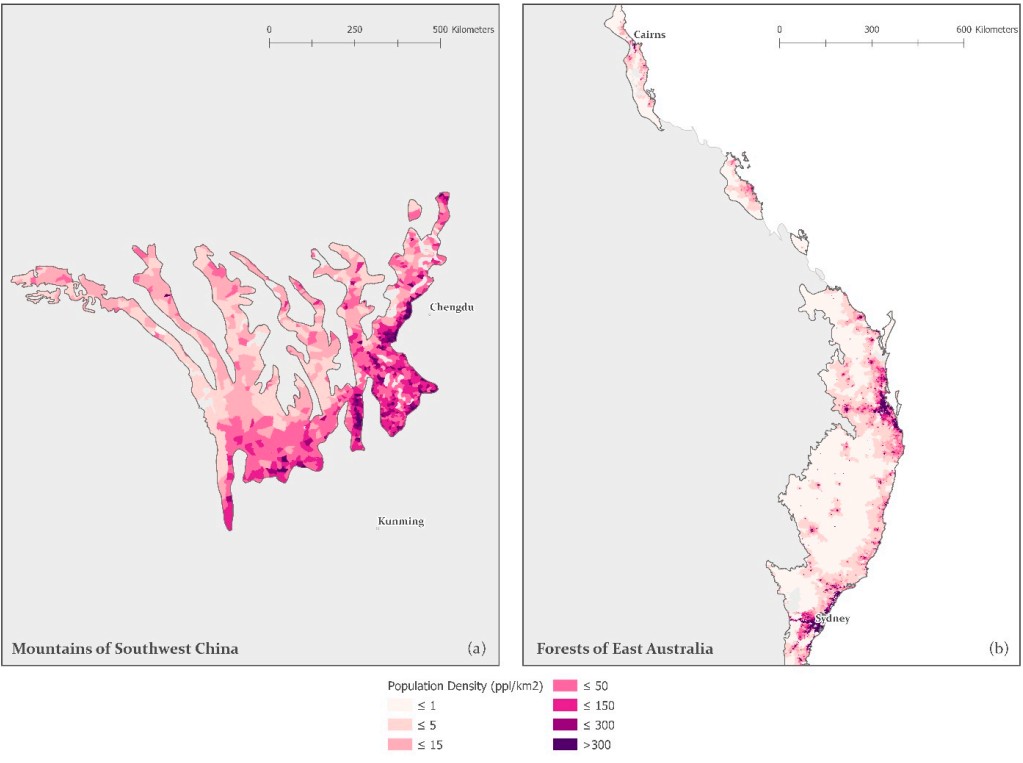

**Figure 4.** Maps showing the population density of two biodiversity hotspots, the Mountains of Southwest China (11.3% protected area in 2015) (**a**) and the Forests of East Australia (23.1% protected area in 2015) (**b**). Both hotspots have similar average population densities (38 and 41 ppl/km$^2$, respectively) and total areas (262,129 and 254,388 km$^2$, respectively), but the patterns of population distribution differ.

Despite these limitations, population density is generally accepted as an indicator for assessing broad trends over large areas and can be considered a better measure of environmental impact than total population or population growth [34]. However, both density and growth rates mask the cultural aspects of the populations they represent and the impacts they have on their surrounding environment (e.g., how affluent they are and what technologies they utilize) [25]. Furthermore, understanding of the human population density-biodiversity relationship is biased towards global, coarse scaled studies [34]. However, given that this study is a global, coarse scaled study, assessing human population density as a factor in biodiversity conservation is appropriate. Although it cannot be said definitively that increasing human population density in biodiversity hotspots directly threatens the biodiversity of these areas, there is a general consensus that the presence of human populations is a key factor contributing to the decline and extirpation of native species [1–3]. Furthermore, an increased human presence can hinder conservation efforts and the establishment of new protected areas in a region [34]. Nonetheless, it is important to measure indicators other than population density alone.

*4.2. Wealth Inequality and Conservation in Biodiversity Hotspots*

A potentially worrying trend is the wealth inequality between the countries where the biodiversity hotspots are considered critical or threatened in terms of population density and area protected, in comparison to those deemed low or least concern. As noted by Mittermeier and colleagues [11], one of the major outcomes of the hotspot designations was an increase in global conservation funds (in excess of 1 billion dollars, US) being directed at some of the most ecologically sensitive places on Earth. While progress has been made in a number of hotspots (for example, in the Cape Floristic Region (3) protected areas more than tripled in size between 1995 and the present), about half (18 of 35) hotspots do not meet 17% protected area as set out by Aichi target 11, and only two exceed 50%. With the notable exception of the California Floristic Province (2) (one of the wealthiest areas on the planet), hotspots that have not achieved the 17% protected area target are located in countries that are struggling economically [48], many of which are also dealing with other significant challenges such as war (Irano-Anatolian (16) and Horn of Africa (14)—civil wars in Syria, Somalia and Yemen), famine (Horn of Africa (14)—associated with the aforementioned civil wars), social unrest (Indo-Burma (15) and Philippines (28)), and rapid sea level rise (Sundaland (32) and Polynesia-Micronesia (29) [49]. In contrast, despite being one of the wealthiest places on the planet, the California Floristic Province has failed to protect 17%. One contributing factor may be that this hotspot has the highest growth in population density since 1995 and the fourth highest average population density. However, 17% protection has been met in the four other hotspots with equivalent (Guinean Forests of West Africa (12) and Caribbean Islands (4)) or higher (Japan (17) and Western Ghats and Sri Lanka (36)) average population densities. The only other hotspot with higher population density in which the 17% target has not been met is the Philippines (2), which, as noted, is struggling with significant economic and social challenges. Half of the biodiversity hotspots that do not meet the 17% protected area target have average population densities above the global average, a factor that will further complicate conservation efforts in those regions [34].

Wealthy nations, such as the United States, in which the California Floristic Province is located, need to step up protection efforts for global biodiversity hotspots located within their boundaries. Those that are signatory to the Convention on Biodiversity should at a minimum follow through on their commitments to Aichi target 11. Beyond these internal commitments, however, relatively wealthy nations should recognize their responsibilities to other global hotspots in terms of world leadership and stewardship in general, but also in response to their indirect role in the loss of natural habitat in the hotspots, such as through their consumption of agricultural and other products produced in those regions and exported through global trade [37]. It is unrealistic, and unethical in terms of international equity, to assume that conservation targets will be met in economically and socially challenged regions without extensive support from the global community.

*4.3. Towards 2020 and Aichi Target 11*

Over the 30 years since Myers [12] first introduced the concept of biodiversity hotspots there has been significant progress in conservation within the hotspots, but threats to their conservation have also continued. Looking towards 2020 and beyond, more resources will need to be directed at biodiversity hotspots. It is important that conservation efforts be directed at places that are vulnerable due to high human presence, in addition to regions where fewer people live, though these less-populated places may also be highly threatened by human activities such as resource extraction. As Cincotta et al. [25] note, biodiversity hotspots with low population density may also be highly threatened by extensive and intensive resource extraction, such as through disturbances related to 'over-logging, burning, grazing, mining and commercial hunting that have extracted or degraded natural resources, abetted biological invasion or polluted soil and water resources' [50]. Many hotspots are at risk from land conversions to agriculture, the effects of which are further compounded by international trade and the consumption of the products in wealthy nations [37]. Consideration of these various factors can result in different estimations of threat.

Of the 10 hotspots that Weinzettel et al. [37] identified as facing the greatest threats from agriculture, only four fell into our high threat category (Irano-Anatolian (16), Maputaland-Pondoland-Albany (20), Sundaland (32) and Wallacea (35)), and two into our low threat category (New Zealand (26) and Succulent Karoo (31)). The remaining four (Cape Floristic Region (3), Mediterranean basin (21), Mountains of Southwest China (24) and Western Ghat and SriLank (36)) had human populations densities higher than the 1995 global average, but also had 17% or more in protected area, thus reducing their threat level in our assessment. Such variations point to the importance of considering multiple factors in identifying threats so as to respond with various forms of conservation initiatives to address fundamental and proximate causes, local and global mechanisms, and reactive and proactive approaches. Furthermore, that two different approaches identified four hotspots in common as highly threatened lends confidence to the notion that significant and immediate conservation attention should be brought to these hotspots.

The conservation targets set out in Aichi Target 11 are political in nature, and fall far below the ecologically-based 25–75% protection estimated by scientific studies (for reviews, see for example, [31–33]. The 17% target should therefore be regarded as a stepping stone towards more wide-spread protection. Current levels of protection, even in regions that meet Aichi Target 11, are woefully inadequate. This is especially true in the most biodiverse and/or threatened places on Earth, such as the biodiversity hotspots, where best practices would call for higher than average percentages of protection. Therefore, we echo the calls made by Baillie and Zhang [30], Wilson [51], Locke [52], and others for increased biodiversity conservation targets—that current conservation targets are too low, especially in the most vulnerable regions of the planet. Targets for biodiversity conservation should err on the side of caution, and given the irreplaceable nature of the hotspots, it is not unreasonable to call for 100% of their remaining intact ecosystems to be protected. The consequences of meeting targets that are too low are far greater than meeting those that are too high. On the other hand, a strict focus on percentage-based targets alone can have many unintended and perverse consequences [53], and thus efforts must take into account the needs of local peoples, careful siting and design of protected areas, and provisions for their effective governance and management, to maximize conservation impact.

A factor which was not examined in this study, but which will be important in future conservation planning in the hotspots, is ecological connectivity and permeability, the degree to which landscapes are able to facilitate species movement between them [54,55]. Similar to the different patterns of population density and protected areas in biodiversity hotspots (Figure 5), there are also different degrees to which species movement can be facilitated on the landscape. Although direct connectivity between protected areas is important, the permeability of the matrix within hotspots must also be considered to facilitate the ecosystem processes that keep the hotspots functional [55,56]. Considerations of permeability will be of importance where there is a high degree of human presence on the landscape, as ecological function is especially vulnerable in such landscapes [57].

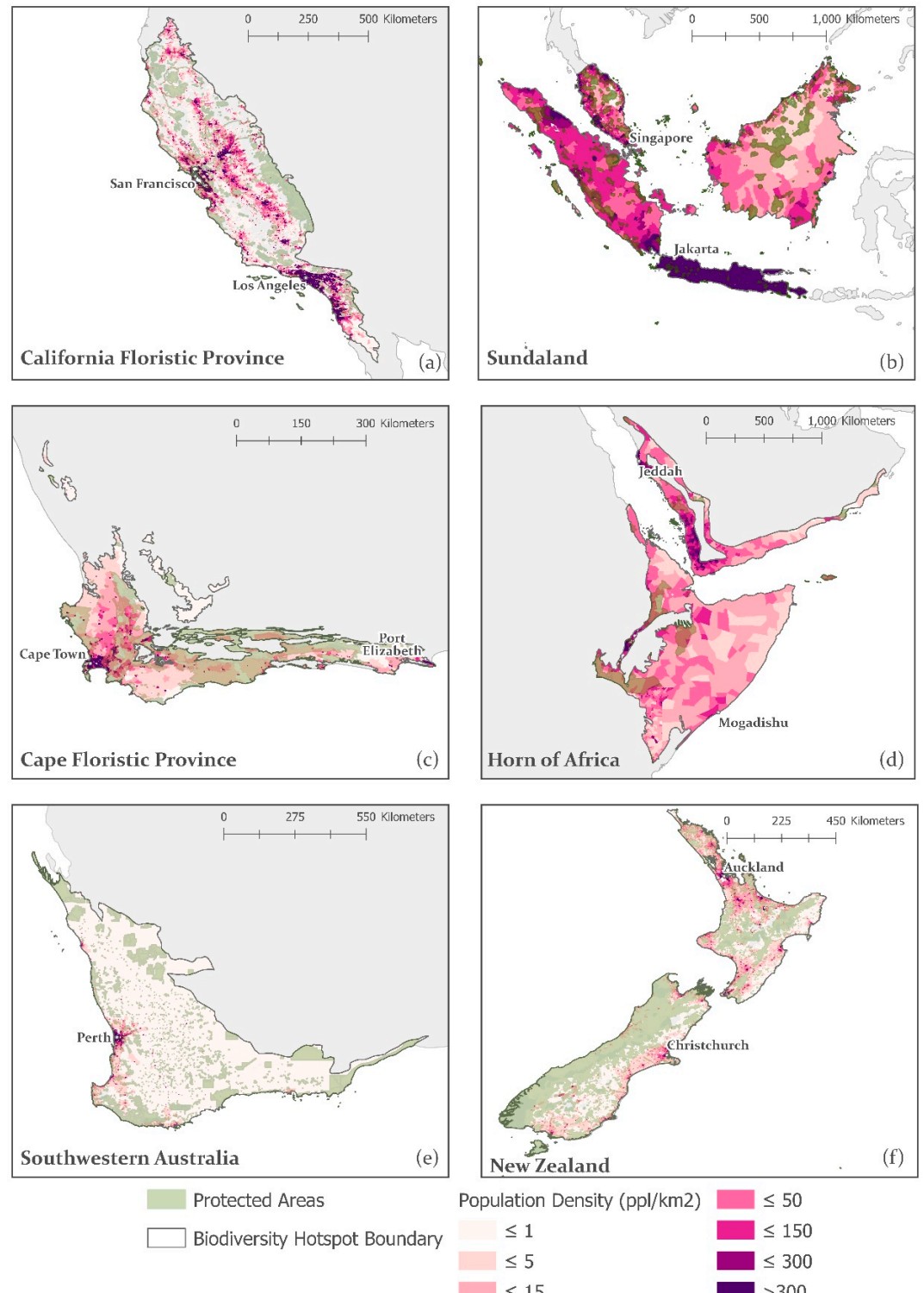

**Figure 5.** Patterns of protected area and population density in biodiversity hotspots: (**a**) California Floristic Province (15.3% protected area in 2015), (**b**) Sundaland (13.6% protected area in 2015), (**c**) Cape Floristic Province (47.4% protected area in 2015), (**d**) Horn of Africa (8.8% protected area in 2015), (**e**) Southwestern Australia (17.1% protected area in 2015), and (**f**) New Zealand (32.6% protected area in 2015).

Taken together, the various patterns of population density and protected areas offer a picture of the current state of conservation in each hotspot and provide perspective on the challenges that lay ahead for maintaining and/or restoring ecological connectivity. For example, in the California Floristic

Province (2) (Figure 5a), the majority of protected areas are located away from areas of high population density (with the exception of a few protected areas around the San Francisco Bay Area), but they are not well connected to each other. In New Zealand (26) (Figure 5f) the protected areas are similarly located away from where people live, but they are much more connected than in California, particularly on the South Island. Conversely, in the Cape Floristic Region (3) (Figure 5c) there is a reasonably well-connected system of protected areas that runs through the city of Cape Town, effectively bisecting the densest part of the city.

Immense challenges lie ahead for biodiversity hotspots where there is little protected area and high population density, such as Sundaland (32) (Figure 5b) and the Horn of Africa (14) (Figure 5d). In these regions there is often a high reliance on natural resources by the local peoples. It is likely that a simple expansion of protected area will be insufficient in all cultural contexts, particularly in areas where illegal logging and poaching remain a major problem. Conservation efforts should, therefore, entail other culturally-appropriate measures to safeguard biodiversity [35,58]. Conservation in areas where people live and work is likely to look different in different cultural contexts.

### 4.4. Data Limitations

There are numerous acknowledged issues with the World Database of Protected Areas (WDPA) that limit the accuracy of this analysis. One issue is spatial accuracy—the data in the WDPA is compiled by the UNEP in collaboration with a wide range of government and NGO partners. A wide range of sources means that there are a variety of different scales and techniques used to gather the data in the WDPA [39]. As discussed in the methods section, roughly 9% of protected areas in the WDPA are recorded as points rather than polygons. Excluding these areas would, therefore, likely result in a substantial underestimate of protected areas. In order to include these protected areas, buffers were created in accordance with the recorded area for each protected area [39]. While this does not account for the shape and extent of protected areas, it does allow for general spatial/areal analyses, sufficient for this analysis. Finally, there are many protected areas which overlap in the WDPA, an issue that is augmented with the addition of buffers on the point data [39]. As described in the methods section, this issue was resolved by running the dissolve tool on the data in ArcGIS Pro (ESRI: Redlands, California, United States) to eliminate overlap.

### 4.5. Future Conservation Planning

Factors other than human population density and percent protected area contribute to the status of biodiversity and its conservation and thus must be considered when setting conservation objectives, including (i) threats such as habitat loss, climate change, overexploitation, invasive species [59], and (ii) the effectiveness of protected areas, measured as the strength of protection and size and/or connectivity of the protected areas network. It should also be noted that setting conservation priorities on the basis of biodiversity hotspots defined as those rich in endemic species and threatened by anthropogenic influences is a relatively reactive approach, as they are by definition areas that are already degraded or threatened. Therefore, an approach that focuses on critical biodiversity hotspots should be balanced by a proactive approach that incorporates some of the least-threatened highly diverse regions, in order to protect the best of what's left of the world's biodiversity while human pressures and interests in those areas remain less intense.

Although there are a handful of protected areas around the world within densely populated areas (such as Table Mountain National Park in the Cape Floristic Region (3)), they are not the norm, and the general paradigm is that conservation is an act that happens away from where people live. However, the reality is that hundreds of millions of people live in some of the most biologically diverse places on Earth. A fundamental shift in what we think conservation is supposed to look like is needed. Conservation is not simply something that happens 'over there', on lands away from humans, but rather it is possible and, as these results indicate, necessary, to meet biodiversity conservation targets where many people live and work, as well as in more remote regions. To adequately safe

guard biodiversity we need to adopt new approaches to conservation that recognize that we are part of biodiversity and that areas of high human population density do not need to be places of conflict with nature, but rather they should be places of coexistence [45–47]. To achieve this, collaboration and participatory action in conservation planning will be crucial, especially across multiple scales [60,61]. A framework for how this could be carried out on the ground are the landscape conservation cooperatives (LCC) in the United States, which show promise that large scale collaboration across governments, NGOs and others is possible. LCCs aim to connect a variety of actors through a bottom up approach to engagement in systematic conservation planning at regional scales [61] The model could prove to be useful for safeguarding biodiversity on large scales all over the world, including in the hotspots.

## 5. Conclusions

This study examined two of many factors that affect biodiversity, human population density and percent protected area. Although significant progress has been made through conservation efforts in biodiversity hotspots over the last twenty years, more is needed. By 2015, the Aichi target 11 goal of 17% protection of terrestrial lands and inland waters had been met in 19 of the 36 hotspots. Only two hotspots had achieved the more-ecologically-realistic target of 50%. The majority of hotspots that did not meet the 17% target are in economically disadvantaged nations. Efforts to meet and exceed these targets will require financial support and other resources from the global community. Compounding the challenge of these future efforts will be increasing human presence in the biodiversity hotspots. Hundreds of millions of people currently live in these areas, and ongoing increases in their populations are putting more pressure on already strained ecosystems. To meet conservation targets in the biodiversity hotspots, an approach that recognizes that humans are part of nature and that areas of high population density can become areas where people live in coexistence with the natural world is necessary.

**Author Contributions:** C.C. designed and conducted the analysis in GIS, performed statistical analyses and drafted the manuscript. K.F.B. contributed to the analysis of results and the writing and editing of the manuscript.

**Funding:** This work was supported by the Nova Scotia Graduate Scholarship, awarded to C.C.

**Acknowledgments:** Thank you to the GIS Centre at Dalhousie University for their support of this work. We would also like to thank the Center for International Earth Science Informational Network (CIESN), the United Nations Environmental Programme (UNEP) and the International Union for Conservation of Nature (IUCN) for providing their data open access and free of charge. Without organizations providing easily accessible data, research like this would not be possible. Thank you to the four reviewers who provided valuable constructive feedback on an earlier draft of this paper.

**Conflicts of Interest:** The authors declare no conflict of interest.

# Appendix A

**Table A1.** Population Density Data for each Biodiversity Hotspot.

| Biodiversity Hotspot | Population Density, 1995 (ppl/km$^2$) | Population Density, 2015 (ppl/km$^2$) | Change in Population Density, 1995–2015 (ppl/km$^2$) | Change in Population Density, 1995–2015 (%) | Predicted Population Density, 2020 (ppl/km$^2$) | Predicted Change in Population Density, 2015–2020 (ppl/km$^2$) | Predicted Change in Population Density, 2015–2020 (%) |
|---|---|---|---|---|---|---|---|
| GLOBAL | 38 | 56 | 18 | 47.37 | 61 | 5 | 8.93 |
| GLOBAL (BIODIVERSITY HOTSPOTS ONLY) | 76 | 103 | 27 | 35.53 | 112 | 9 | 8.74 |
| Atlantic Forest (1) | 77 | 98 | 21 | 27.3 | 104 | 6 | 6.1 |
| California Floristic Province (2) | 108 | 202 | 94 | 87.0 | 213 | 11 | 5.4 |
| Cape Floristic Region (3) | 48 | 80 | 32 | 66.7 | 91 | 11 | 13.8 |
| Caribbean Islands (4) | 150 | 189 | 39 | 26.0 | 208 | 19 | 10.1 |
| Caucasus (5) | 67 | 73 | 6 | 9.0 | 75 | 2 | 2.7 |
| Cerrado (6) | 11 | 16 | 5 | 45.5 | 17 | 1 | 6.3 |
| Chilean Winter Rainfall and Valdivian Forests (7) | 29 | 42 | 13 | 44.8 | 47 | 5 | 11.9 |
| Coastal Forests of Eastern Africa (8) | 49 | 73 | 24 | 49.0 | 85 | 12 | 16.4 |
| East Melanesian Islands (9) | 12 | 19 | 7 | 58.3 | 22 | 3 | 15.8 |
| Eastern Afromontane (10) | 84 | 143 | 59 | 70.2 | 166 | 23 | 16.1 |
| Forests of East Australia (11) | 33 | 41 | 8 | 24.2 | 41 | 0 | 0.0 |
| Guinean Forests of West Africa (12) | 111 | 187 | 76 | 68.5 | 216 | 29 | 15.5 |
| Himalaya (13) | 111 | 169 | 58 | 52.3 | 197 | 28 | 16.6 |
| Horn of Africa (14) | 16 | 29 | 13 | 81.3 | 35 | 6 | 20.7 |
| Indo-Burma (15) | 128 | 170 | 42 | 32.8 | 183 | 13 | 7.6 |
| Irano-Anatolian (16) | 60 | 69 | 9 | 15.0 | 73 | 4 | 5.8 |
| Japan (17) | 330 | 336 | 6 | 1.8 | 337 | 1 | 0.3 |
| Madagascar and the Indian Ocean Islands (18) | 27 | 44 | 17 | 63.0 | 52 | 8 | 18.2 |
| Madrean Pine-Oak Woodlands (19) | 45 | 45 | 0 | 0.0 | 50 | 5 | 11.1 |
| Maputaland-Pondoland-Albany (20) | 65 | 85 | 20 | 30.8 | 90 | 5 | 5.9 |
| Mediterranean Basin (21) | 100 | 133 | 33 | 33.0 | 143 | 10 | 7.5 |
| Mesoamerica (22) | 60 | 89 | 29 | 48.3 | 99 | 10 | 11.2 |
| Mountains of Central Asia (23) | 39 | 55 | 16 | 41.0 | 60 | 5 | 9.1 |
| Mountains of Southwest China (24) | 39 | 38 | −1 | −2.6 | 40 | 2 | 5.3 |
| New Caledonia (25) | 11 | 13 | 2 | 18.2 | 15 | 2 | 15.4 |
| New Zealand (26) | 13 | 18 | 5 | 38.5 | 19 | 1 | 5.6 |
| North American Coastal Plain (27) | 56 | 87 | 31 | 55.4 | 97 | 10 | 11.5 |
| Philippines (28) | 242 | 345 | 103 | 42.6 | 378 | 33 | 9.6 |
| Polynesia-Micronesia (29) | 63 | 89 | 26 | 41.3 | 94 | 5 | 5.6 |
| Southwest Australia (30) | 4 | 6 | 2 | 50.0 | 6 | 0 | 0.0 |
| Succulent Karoo (31) | 4 | 3 | −1 | −25.0 | 4 | 1 | 33.3 |
| Sundaland (32) | 130 | 172 | 42 | 32.3 | 187 | 15 | 8.7 |
| Tropical Andes (33) | 34 | 46 | 12 | 35.3 | 51 | 5 | 10.9 |
| Tumbes-Choco-Magdalena (34) | 43 | 63 | 20 | 46.5 | 72 | 9 | 14.3 |
| Wallacea (35) | 71 | 100 | 29 | 40.8 | 112 | 12 | 12.0 |
| Western Ghats and Sri Lanka (36) | 276 | 335 | 59 | 21.4 | 354 | 19 | 5.7 |

**Table A2.** Protected Area Data for Biodiversity Hotspots.

| Biodiversity Hotspot | Protected Area, 1995 (%) | 1995 Protected Area +/− 17% | Protected Area, 2015 (%) | 2015 Protected Area +/− 17% | Change in Protected Area, 1995–2015 (%) |
|---|---|---|---|---|---|
| GLOBAL | 9.8 | −7.2 | 15.4 | −1.6 | 5.6 |
| GLOBAL (HOTSPOTS ONLY) | 10.64 | −6.36 | 24.22 | +7.2 | 13.57 |
| Atlantic Forest (1) | 16.53 | −0.47 | 41.60 | +24.60 | 25.08 |
| California Floristic Province (2) | 13.61 | −3.39 | 15.34 | −1.66 | 1.74 |
| Cape Floristic Region (3) | 13.74 | −3.26 | 47.42 | +30.42 | 33.69 |
| Caribbean Islands (4) | 5.15 | −11.85 | 31.79 | +14.79 | 26.64 |
| Caucasus (5) | 8.30 | −8.70 | 10.09 | −6.91 | 1.78 |
| Cerrado (6) | 17.02 | +0.02 | 53.88 | +36.88 | 36.86 |
| Chilean Winter Rainfall and Valdivian Forests (7) | 12.36 | −4.64 | 35.97 | +18.97 | 23.61 |
| Coastal Forests of Eastern Africa (8) | 11.68 | −5.32 | 21.92 | +4.92 | 10.24 |
| East Melanesian Islands (9) | 1.46 | −15.54 | 2.19 | −14.81 | 0.73 |
| Eastern Afromontane (10) | 18.77 | +1.77 | 19.63 | +2.63 | 0.86 |
| Forests of East Australia (11) | 6.50 | −10.50 | 23.07 | +6.07 | 16.57 |
| Guinean Forests of West Africa (12) | 15.31 | −1.69 | 33.85 | +16.85 | 18.54 |
| Himalaya (13) | 9.73 | −7.27 | 16.88 | −0.12 | 7.15 |
| Horn of Africa (14) | 8.39 | −8.61 | 8.85 | −8.15 | 0.46 |
| Indo-Burma (15) | 7.49 | −9.51 | 13.32 | −3.68 | 5.83 |
| Irano-Anatolian (16) | 4.27 | −12.73 | 5.01 | −11.99 | 0.74 |
| Japan (17) | 27.29 | +10.29 | 29.42 | +12.42 | 2.14 |
| Madagascar and the Indian Ocean Islands (18) | 2.78 | −14.22 | 11.69 | −5.31 | 8.92 |
| Madrean Pine-Oak Woodlands (19) | 2.75 | −14.25 | 15.12 | −1.88 | 12.37 |
| Maputaland-Pondoland-Albany (20) | 7.60 | −9.40 | 10.03 | −6.97 | 2.43 |
| Mediterranean Basin (21) | 7.08 | −9.92 | 21.59 | +4.59 | 14.51 |
| Mesoamerica (22) | 15.55 | −1.45 | 35.02 | +18.02 | 19.46 |
| Mountains of Central Asia (23) | 5.34 | −11.66 | 13.99 | −3.01 | 8.65 |
| Mountains of Southwest China (24) | 0.95 | −16.05 | 11.26 | −5.74 | 10.31 |
| New Caledonia (25) | 22.41 | +5.41 | 57.74 | +40.74 | 35.33 |
| New Zealand (26) | 25.26 | +8.26 | 32.57 | +15.57 | 7.31 |
| North American Coastal Plain (27) | 5.17 | −11.83 | 6.89 | −10.11 | 1.71 |
| Philippines (28) | 8.86 | −8.14 | 15.44 | −1.56 | 6.58 |
| Polynesia-Micronesia (29) | 9.14 | −7.86 | 13.96 | −3.04 | 4.82 |
| Southwest Australia (30) | 15.10 | −1.90 | 17.06 | +0.06 | 1.96 |
| Succulent Karoo (31) | 3.08 | −13.92 | 31.01 | +14.01 | 27.93 |
| Sundaland (32) | 12.79 | −4.21 | 13.58 | −3.42 | 0.79 |
| Tropical Andes (33) | 15.46 | −1.54 | 46.72 | +29.72 | 31.25 |
| Tumbes-Choco-Magdalena (34) | 10.19 | −6.81 | 23.15 | +6.15 | 12.97 |
| Wallacea (35) | 9.88 | −7.12 | 10.90 | −6.10 | 1.02 |
| Western Ghats and Sri Lanka (36) | 17.42 | +0.42 | 24.35 | +7.35 | 6.93 |

**Table A3.** Correlation Coefficients for Population Density and Protected Area in Biodiversity Hotspots.

| | 1995 Population Density (ppl/km²) | 2015 Population Density (ppl/km²) | 2020 Predicted Population Density (ppl/km²) | Population Density Change (1995–2015) | Population Density Change (2015–2020) | 1995 Protected Areas (%) | 2018 Protected Areas (%) | Protected Areas Change (1995–2018) |
|---|---|---|---|---|---|---|---|---|
| 1995 Population Density (ppl/km²) | 1.00 | | | | | | | |
| 2015 Population Density (ppl/km²) | 0.97 | 1.00 | | | | | | |
| 2020 Predicted Population Density (ppl/km²) | 0.96 | 1.00 | 1.00 | | | | | |
| Population Density Change (1995–2015) | 0.55 | 0.73 | 0.77 | 1.00 | | | | |
| Population Density Change (2015–2020) | 0.51 | 0.67 | 0.72 | 0.88 | 1.00 | | | |
| 1995 Protected Areas (%) | 0.30 | 0.27 | 0.26 | 0.10 | 0.03 | 1.00 | | |
| 2018 Protected Areas (%) | −0.07 | −0.09 | −0.09 | −0.12 | −0.09 | 0.61 | 1.00 | |
| Protected Areas Change (1995–2018) | −0.26 | −0.27 | −0.26 | −0.20 | −0.12 | 0.19 | 0.89 | 1.00 |

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
