# Peer review of "Changes in Human Population Density and Protected Areas in Terrestrial Global Biodiversity Hotspots, 1995–2015"

_land, doi:10.3390/land7040136_

Round 1

Reviewer 1 Report

Thank you for a well-written, straightforward, and easy to read paper. When I first skimmed it, I wondered if human population density + protected area was a sufficient proxy for risk.... between the time I first skimmed it and this second read, a paper came out in Frontiers in Ecology and the Environment which also looks at risks to these 36 hotspots in terms of land use (NPP appropriation as a proxy), specifically agriculture and pasture: Weinzettel et al. 2018, "Human footprint in biodiversity hotspots", Frontiers in Ecology and the Environment 16(8):447-452. 

It would really benefit your paper to spend considerable space in the Discussion section placing your results in context with this paper by Weinzettel et al.... combined, the two papers examine two major risks (land use and human population) along with % protected area (which you include but Weinzettel et al don't). See lines 347-341... this is likely the best spot to incorporate this Weinzettel paper (when you discuss Cincotta et al.)

Figure 4. Please add to the caption the % protected area in 2015 for those two hotspots....

The Results section is a bit tedious and repetitious with the Appendices. You can probably tighten this up a bit and focus on the key points or areas.

Lines 320-325: The California Floristic Region is an important exception.... rapid population growth, lack of hitting even the 17% protected area target despite being one of the wealthiest places on the planet. I'd recommend spending at least a sentence or two reviewing this exception. (You do this to some extent in lines 365-371, but talking about connectivity of protected areas.)

Lines 358-371: Yes connectivity of protected areas is important, but so is the permeability of the landscape these protected areas sit in. See for example: Theobald et al. 2012. Connecting natural landscapes using a landscape permeability model to prioritize conservation activities in the United States. Conservation Letters 5(2): 123-133 and papers cited on permeability therein.

Lines 410-420: Might be good to cite some papers which talk about ways to go about this proactive, coexistence approach in a) urban areas and b) agricultural areas... e.g., papers on "land sharing" approach.

Reviewer 2 Report

General remarks

In a general way, the paper reflects a very interesting research in order to point out the several threats to biodiversity, but in my opinion, and in the present way, the paper is so much descriptive, and some more "deep" research should be done.

The first part of the paper, putting in relationship human densities and hotspots is very useful, a really had work, but it is just a kind of descriptive work. The author's comments in relationship with other issues are very important, and they should do a bit more research in this way, in order to attach a more feasible scientific approach to the paper: e.g. The authors talk about the crucial issues in interaction human densities-natural preservation like the importance of how human densities are distributed through the landscapes, on a sharing way, on a more sparing one. Or the authors point out, and do some very useful work-scientific sound, feasible science about it: the relevance of landscape/habitat fragmentation. The paper should presents   a bit more work/research in relationships with the issues reflected above, because in in the actual way it is characterized by a  more descriptive  results and more theoretical, logical discussion and conclusions  than a real ones.

In a particular way, some sections should be rewriting, perhaps in a Table format, because the text presents a lot of data and information which are "inside of the shadows of descriptive writing", e.g. section 3.3., and so one. Showing the figures in a Table format may be more useful to understand the results more that show it in a "writing form". Of course, you should be care about do not repeat information through the Text, Tables, and Figures.

Specifically remarks

e.g. Section 2.3. A more details should be show about statistical procedures, e.g. which statistical-test was used in orderr to  run regression  between population density and a percentage of protected areas.

e.g. Section 4.5. Future research.   A too general reflection that does not added relevant information to the research objective and should be remove it.

In this section 4.5., some aspects should be move to Conclusions section, especially those in relation to your research finds.

e.g. 5. Conclusions section should show, really, your research finds, not a general approach to possible consequences and or global challenges in relationship with your research topic.

Reviewer 3 Report

This is an interesting paper on the vulnerability of the global biodiversity hotspots. I made a few minor comments:

- In the introduction, authors have described the current situation of biodiversity hotspots conservation, but, in my opinion, this section and / or discussion should further discuss previous studies about the prioritization of sites of high conservation value lack of efficient instruments for coordinating conservation initiatives and about the role of collaboration in nature conservation.

- Lines 95 – 104 are the same with 211 – 223 presenting the info about the represented hotspots and then repeat again in the figure captions 2 and 3. This basically means that 274 words repeat in the MS  4 times – over 1000 words. I consider that the authors should find a solution to reduce this repletion – once is enough and after just make reference to the legend of the hotspots. Given the reduction of the text the authors could better discuss and highlight their discoveries, contributions, and extensions.

- Just a suggestion in Figure 1. The numbers of the hotspots should have a different font for a better visualization of the map.

Reviewer 4 Report

This is a helpful progress report on the biodiversity hotspots concept and its application.  One problem is that the "hotspots" are mostly rather large, with population density widely variable within them.

Round 2

Reviewer 2 Report

Changes were done in a properly way